# HbA1c and Glucose Management Indicator Discordance Associated with Obesity and Type 2 Diabetes in Intermittent Scanning Glucose Monitoring System

**DOI:** 10.3390/bios12050288

**Published:** 2022-04-29

**Authors:** Paul Fellinger, Karin Rodewald, Moritz Ferch, Bianca Itariu, Alexandra Kautzky-Willer, Yvonne Winhofer

**Affiliations:** Division of Endocrinology and Metabolism, Internal Medicine III, Waehringer Guertel 18–20, 1090 Vienna, Austria; paul.fellinger@meduniwien.ac.at (P.F.); karin@rodewald.cc (K.R.); moritz.ferch@meduniwien.ac.at (M.F.); bianca.itariu@meduniwien.ac.at (B.I.); alexandra.kautzky-willer@meduniwien.ac.at (A.K.-W.)

**Keywords:** continuous glucose monitoring, GMI, HbA1c, type 2 diabetes, overweight

## Abstract

Glucose management indicator (GMI) is frequently used as a substitute for HbA1c, especially when using telemedicine. Discordances between GMI and HbA1c were previously mostly reported in populations with type 1 diabetes (T1DM) using real-time CGM. Our aim was to investigate the accordance between GMI and HbA1c in patients with diabetes using intermittent scanning CGM (isCGM). In this retrospective cross-sectional study, patients with diabetes who used isCGM >70% of the time of the investigated time periods were included. GMI of four different time spans (between 14 and 30 days), covering a period of 3 months, reflected by the HbA1c, were investigated. The influence of clinical- and isCGM-derived parameters on the discordance was assessed. We included 278 patients (55% T1DM; 33% type 2 diabetes (T2DM)) with a mean HbA1c of 7.63%. The mean GMI of the four time periods was between 7.19% and 7.25%. On average, the absolute deviation between the four calculated GMIs and HbA1c ranged from 0.6% to 0.65%. The discordance was greater with increased BMI, a diagnosis of T2DM, and a greater difference between the most recent GMI and GMI assessed 8 to 10 weeks prior to HbA1c assessment. Our data shows that, especially in patients with increased BMI and T2DM, this difference is more pronounced and should therefore be considered when making therapeutic decisions.

## 1. Introduction

The introduction of continuous glucose monitoring (CGM) in recent years improved the management of diabetes significantly [1]. By using CGM systems, a variety of new parameters, such as “time in range” (TiR), “time below range” (TbR), “time above range” (TaR), and the coefficient of variation (CV) were established to describe glycaemic control of patients [2]. To estimate HbA1c, the current gold standard for glycaemic control, the parameter glucose management indicator (GMI), based on average glucose measurements, was introduced [3]. Before that, the so-called eHbA1c was used; however, this term was abandoned due to confusion with the real HbA1c assessed by blood sample. The current understanding is that sensor data covering 14 days are sufficient to calculate GMI representatively [4]. Through its easy accessibility, patients often rely on this parameter, especially in the last years when treatment via telemedicine became more important during the COVID-19 pandemic [5]. However, recent publications show that there are significant deviations between GMI and HbA1c, especially when glycaemic control is worse [6,7]. A recent review also suggested to identify the glycation rates and status of patients using the difference between GMI and HbA1c, implying that a large difference between the measurement implies low glycation rates and vice versa [8].

Most studies so far predominantly used data from patients with type 1 diabetes, which might make the formula susceptible to greater deviations in patients with other forms of diabetes [3,6,9]. Furthermore, studies primarily use data from real-time CGM systems (rtCGM), which transmit glucose values continuously every few minutes. On the other hand, intermittent scanning systems (isCGM), which transmit glucose measurements when the transmitter is scanned by the patient, use data obtained every 15 min. As intermittent scanning (isCGM) is currently the most popular and frequently used system in all forms of diabetes, data on the reliability of the GMI given are necessary [1,10].

Furthermore, data on the right time of GMI assessment within the 3 months, which are reflected by the HbA1c, are of interest.

The aim of this study was to investigate whether GMI is a robust parameter to predict HbA1c and glycaemic control in a real-world setting of patients using Freestyle Libre (Abbott Diabetes Care Inc., Chicago, IL, United States) and to determine factors that are associated with the often observed significant deviations between the measurements.

## 2. Materials and Methods

In this cross-sectional study, we retrospectively included 278 patients treated in the diabetes outpatient clinic of the General Hospital of Vienna between 2019 and 2021 and who were using Freestyle Libre systems. Patients were included when we could obtain CGM data representing four different time spans, namely the periods of 14 days and 30 days, as well as the time spans between week 6 and week 4 and between week 10 and week 8, prior to the HbA1c measurement (see Figure 1). Within those time spans, active CGM time had to be at least 70% or higher. For the time period of 14 days prior to the HbA1c measurement, we obtained the parameters TiR, TaR, TbR, and CV as provided by the Diasend/Glooko platform (Diasend AB, Göteborg, Sweden/Glooko Inc., Mountain View, CA, USA) as well as the LibreView platform (Abbott Diabetes Care Inc., Chicago, IL, United States). For the three other time spans, we retrieved average glucose and GMI only.

GMI was calculated using the formula published by Bergenstal et al. [GMI (%) = 3.31 + 0.02392 × CGM mean glucose (mg/dL)] [3]. HbA1c was measured by high-performance liquid chromatography. We included patients with all diabetes entities, apart from gestational diabetes. For being included, diabetes duration had to be at least 4 months. Patients affected by haemoglobinopathies known to influence HbA1c were excluded [11].

### Statistical Analysis

Continuous variables are presented as mean ± standard deviation and as absolute numbers as well as relative percentages. To analyse agreement between the two different methods, namely HbA1c and GMI, we performed a Bland–Altman plot. The average of the two measurements (on the x-axis) was plotted against the difference between HbA1c and GMI (y-axis) to show the average difference between the measurements. Furthermore, we also calculated the absolute difference between the two measurements by calculating the square root of the number squared.

To evaluate whether the differences are unrelated to the mean, we plotted the mean against the relative difference of the two measurements. To further analyse a potential proportional bias and discover non-uniform differences, we performed a regression analysis between the average between GMI and HbA1c with the differences between the two. We performed these analyses for each of the four time spans. Correlation analyses were performed using spearman coefficient analysis. Linear regression was used to show the correlation of each individual plot. Furthermore, to model the relationship between the absolute difference of GMI1 and HbA1c as a dependent variable and a variety of other parameters as independent variables, a linear regression analysis was carried out. Missing data of parameters are stated if applicable. All reported CIs are at the 95% level. Statistical significance was defined as *p* < 0.05.

This study is approved by the local ethics committee and has been carried out in accordance with the principles of the Declaration of Helsinki as revised in 2008 (classification number 2244/2019). RStudio was used to perform statistical analysis as well as to create graphs (RStudio Team (2021)).

## 3. Results

We included 278 patients with a mean age of 54.8 ± 15.75 years of which 47.5% were female. For further characteristics see Table 1. All patients used CGM device Freestyle Libre 1 (Abbott Diabetes Care Inc., Chicago, IL, United States).

The mean HbA1c was 7.63%, whilst the mean average glucose representing the two weeks prior to the HbA1c measurement was 162.23 mg/dL resulting in a mean GMI of 7.19%. Most of the included patients had type 1 diabetes (55.7%), approximately one-third had type 2 diabetes (32.7%), and around 11% were diagnosed with specific types of diabetes due to other causes, such as cystic fibrosis-related diabetes or pancreas insufficiency. Most patients (82.7%) used multiple daily injection therapy (MDI) with a subcutaneous basal insulin analogue and mealtime rapid acting or ultra-rapid acting insulin analogue. Approximately 10% had continuous subcutaneous insulin infusion pump (CSII) and around a quarter of all patients had Metformin in their therapy. The mean wearing time of the CGM device in the observed time period was 90.12 ± 7.87%.

### 3.1. Quality of Glycaemic Control

As seen in Table 2, only around one-third of our patients reached the target of spending at least 70% of the observed time within the recommended glucose range between 70 mg/dL and 180 mg/dL; almost half of the patients had more than 4% of the observed time glucose values below 70 mg/dL. Average glucose values and, therefore, GMI values as well were similar at all 4 observed time spans.

### 3.2. Analysis of Differences

As expected GMI and HbA1c correlated significantly with each other at each time span (see Table 3).

As seen in Figure 2, the mean difference between HbA1c and GMI was greater than 0.38% at all observed time points, with the largest mean difference when using GMI representing the two weeks prior to HbA1c measurement. The difference between HbA1c and GMI was significantly increasing at all time spans with greater HbA1c and GMI values, respectively. In general, GMI readings were lower than HbA1c values, with the difference increasing with worse glycaemic control. This significant difference is even more pronounced when looking at the absolute value of the difference, calculated as the square root of the difference squared between HbA1c and GMI values, as the mean absolute difference was on average greater than 0.6% in all time spans with a median of 0.49%, respectively (see Figure 3).

The absolute value of the relative difference between the two measurements was calculated and plotted against each other to analyse whether the difference was unrelated to the mean of the two measurements. As seen in Figure 4, the mean difference between the measurement was at least 8.33%. Furthermore, in the regression analysis, the slope remained positive, proving that the difference between the measurements was unrelated to the mean and disproportionately increased with worse glycaemic control.

### 3.3. Regression Analysis

To evaluate which factors could potentially influence this often clinically relevant difference between the two measurements, we performed a linear regression analysis using CGM parameters as well as clinical parameters, which were easily attainable. In Table 4, the included variables to predict the absolute difference between the two measurements are shown. In this linear model, BMI, type 2 diabetes, as well as the absolute difference between the GMI1 and GMI3 were statistically significantly associated with an increased difference between HbA1c and GMI1.

## 4. Discussion

In this retrospective study amongst 278 patients with different types of diabetes, all of whom were using a Freestyle Libre isCGM, we found a significant deviation between GMI and HbA1c. This difference was on average more than 0.6% points across all four observed time spans, thus of clinical relevance. The deviation between HbA1c and GMI was consistent over all time spans within the HbA1c coverage period and remained significant even when analysing relative differences and was therefore unrelated to the mean. This absolute mean deviation was larger than in the original study by Bergenstal et al. but similar to a previously published study, where the median deviation was 0.5% and larger [3,6]. Glycaemic control was comparable with a mean HbA1c von 7.63% to other recent CGM publications where HbA1c was 7.3% and 7.5%, respectively [6,7]. With a mean age of 54.8 years, the patient population was slightly older than in the recent study by Perlman et al., where patients had a mean age of 45.3 years [6].

The observed deviation between GMI and HbA1c might be caused by a sensor bias and a bias towards patients with type 1 diabetes, as GMI was developed using rtCGM systems in patients with type 1 diabetes and might therefore need a separate formula for each different sensor and adjustments for other types of diabetes [3,12]. The Freestyle Libre sensor had a published mean absolute relative difference (MARD) of 13.2% but still had a similar accuracy in a direct comparison study between this sensor and a Dexcom G5 sensor, which was used for the development of the GMI [13,14]. However, in a recent publication by Grimsmann et al., in young patients with type 1 diabetes, a greater deviation between GMI and HbA1c was observed in isCGM systems than in rtCGM [12].

In this current study, GMI did correlate significantly with HbA1c in all four time spans. An increase in sampling duration from 14 days to 30 days only led to a slight but not substantial increase in the correlation coefficient. This furthermore supports previously published results that a sampling duration of ten to fourteen days of CGM data is sufficient to estimate glycaemic control appropriately [3].

Interestingly, the two time spans covering 14 days until 4 and 8 weeks prior had lower correlation coefficients than the two most recent time spans with 0.80 and 0.78, respectively. Almost all patients were on insulin therapy, with most having a regime with multiple daily injections with basal insulin analogues and mealtime rapid-acting insulin analogues. Our study population comprised roughly 50% of patients with type 1 diabetes, and about a third had type 2 diabetes. Therefore, in comparison to other CGM studies, our population comprised a relatively large ratio of patients with type 2 diabetes and a rather large ratio of patients with other types of diabetes, such as cystic fibrosis-related diabetes [6,7].

In the regression analysis, type 2 diabetes was a significant factor associated with an increased difference between GMI and HbA1c. In the study by Perlman et al., where 16% of all patients had type 2 diabetes, this association was not found to be statistically significant [6]. In addition, BMI seems to be a significant factor playing a role in an increased deviation between the two measurements. One hypothesis could be that, especially in overweight patients with type 2 diabetes, the accuracy of CGM measurements is impaired due to a reduction in circulation in subcutaneous fat tissue leading to less diffusion in the sensing area and consequently lower glucose readings [15,16]. Obesity and type 2 diabetes have also been associated with interstitial oedema and increased tissue inflammation, which might also play a role in the greater deviations in this population [17,18]. The observed increased differences, especially in patients with type 2 diabetes, could also imply that patients with type 2 diabetes tend to have a higher glycation status. They might therefore be at higher risk for hypoglycaemic episodes when intensifying treatment than patients with other types of diabetes and a lower glycation status, represented by lower HbA1c than GMI [8]. To evaluate such possible clinical difference, a much larger population sample and a longer observation period would be needed.

Another parameter which showed to be highly significantly associated with the deviation between GMI and HbA1c was the absolute difference between the GMI of the most current time span and the GMI calculated from the time span from 10 to 8 weeks prior to the HbA1c measurement assessment. As HbA1c reflects glycaemic control over roughly 8–12 weeks and is furthermore influenced by the lifespan of red blood cells, changes to diet, therapy regimen, or other factors within this time period are therefore typically not reflected in an adequate change in HbA1c [8,11,19]. We excluded patients with known haemoglobinopathies to exclude major confounders, but unfortunately, we did not have conclusive data regarding kidney function and haemoglobin levels, which were also known to influence HbA1c [6,20].

Usually when looking at CGM readings, data from the last 14 days is used to evaluate glycaemic control. One of the parameters which is frequently used is the coefficient of variation, CV, which represents an index of glycaemic variability [21]. Thus far, however, there is no parameter which shows whether overall glycaemic control is stable within the last weeks and months. Implementing a parameter reflecting long-time glycaemic stability could help identify those patients who are more at risk for greater deviations between HbA1c and GMI on the one side, as well as help the clinician to put possible deviations into context to rely more on HbA1c or GMI for possible therapeutic decisions.

### Limitations/Strengths

One major limitation of this study was that we included a lower number of patients in comparison to other recent studies [6,19]. However, by using only one sensor type and longer observation periods, we make our results more comparable to future studies in that field and introduce less bias. Another strength was that our collective had a large proportion of patients with type 2 diabetes.

## 5. Conclusions

In conclusion, there is a marked and relevant difference between GMI and HbA1c when using isCGM. This deviation between the two measurements seems to be associated with type 2 diabetes, with increasing BMI, as well as with the difference between GMI, measured recently and 8 weeks prior. Therefore, the use of GMI as a substitute for HbA1c should be reconsidered or at least used carefully. Especially when considering therapeutic decisions, other parameters such as TiR, as recommended by the current guidelines, should be used. However, this recommendation needs further confirmation in patients with T2DM and obesity [22]. A useful approach might be to evaluate whether the patient has a greater deviation between the most current GMI and the GMI derived from a time span 10 to 8 weeks prior. This aspect of analysing fluctuations of glycaemic control over a longer time span might help us in the future to understand each patient even better and adapt and monitor our therapy more precisely.

## Figures and Tables

**Figure 1 biosensors-12-00288-f001:**
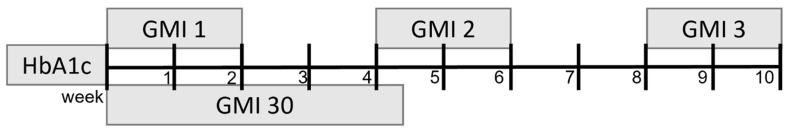
Timeline of GMI assessment in relation to HbA1c measurement.

**Figure 2 biosensors-12-00288-f002:**
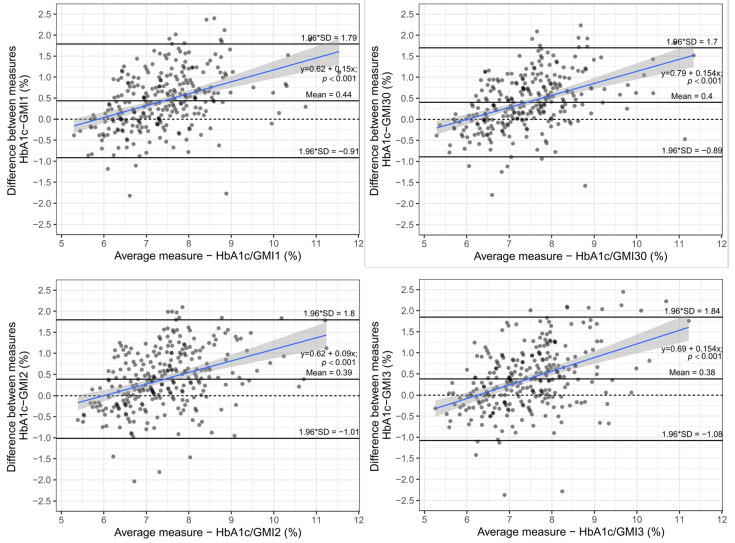
Bland–Altman plot—Difference between HbA1c and GMI measurements vs. average of two measurements.

**Figure 3 biosensors-12-00288-f003:**
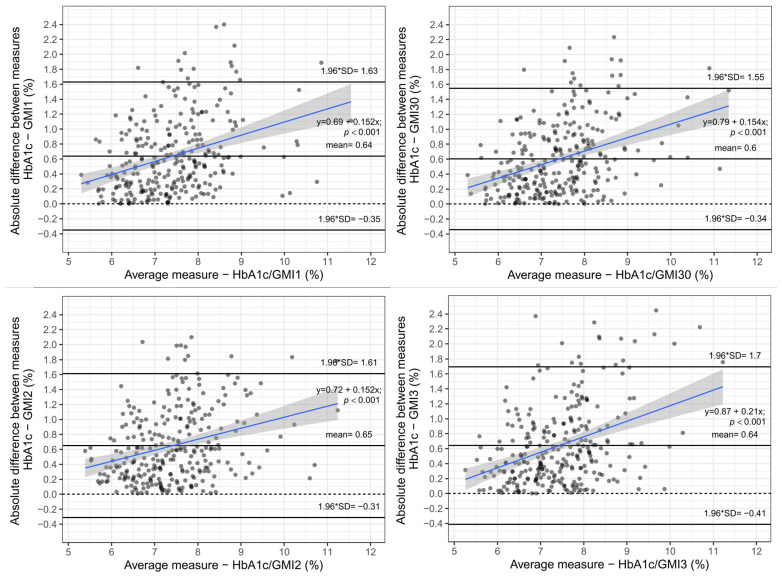
Bland–Altman plot—Absolute value of difference between HbA1c and GMI measurements vs. average of two measurements.

**Figure 4 biosensors-12-00288-f004:**
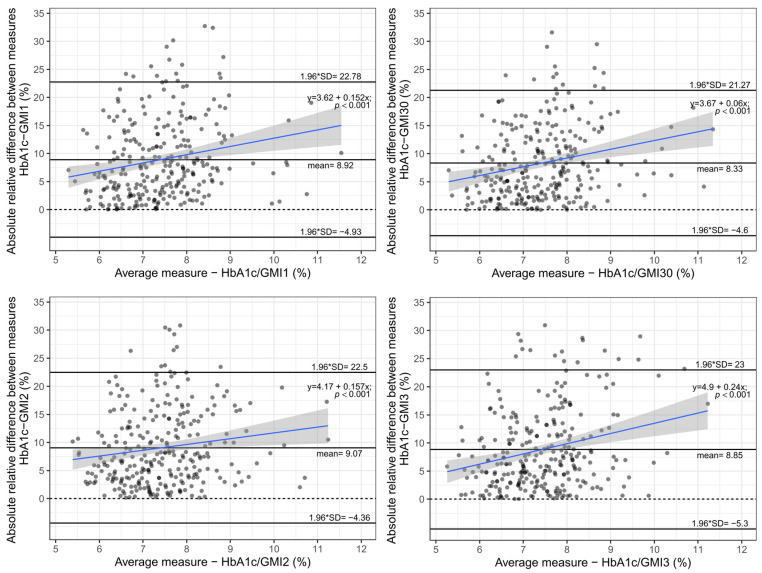
Bland–Altman plot—Absolute value of relative difference between HbA1c and GMI measurements vs. average of two measurements.

**Table 1 biosensors-12-00288-t001:** Baseline characteristics (SA: short acting).

Variable	Mean ± Standard Deviations
N = 278	
Sex (f/m)	47.5%/52.5%
Age (y)	54.8 ± 15.75
BMI (kg/m^2^)	27.94 ± 6.14
RR sys (mmHg)	139.05 ± 20.63
RR dia (mmHg)	83.13 ± 12.24
Type diabetes	
Type 1	155 (55.76%)
Type 2	91 (32.7%)
other types of diabetes	32 (11.5%)
Therapy for diabetes	
no insulin	4 (1.44%)
basal only	13 (4.68%)
basal + prandial	230 (82.73%)
Insulin pump	26 (9.35%)
Only SA insulin	1 (0.36%)
Mix-Insulin	4 (1.44%)
Metformin	71 (25.54%)
HbA1c (mmol/mol)	59.87 ± 13.01
HbA1c (%)	7.63 ± 1.19
HbA1c <7% vs. >7%	82 (29.5%) vs. 196 (70.5%)
duration of diabetes (n194; y)	19.68 ± 14.77 (span: 0.3–66 y)

**Table 2 biosensors-12-00288-t002:** CGM characteristics; (w: week; CV: coefficient of variance).

	Mean ± SD
GMI1 (-d14-d0)	7.19 ± 0.92
Mean glucose (-d14-d0)	162.23 ± 38.39
Time in range (%)	60.71 ± 21.53
TIR >70% vs. <70% of time	99 (35.6%) vs. 179 (64.4%)
Time above range (%)	34.66 ± 22.51
Time below range (%)	4.59 ± 5.14
Time below range >4% of time vs. <4% of time	127 (45.7%) vs. 151 (54.3%)
CV (%)	35 ± 9
GMI30 (-30d-d0)	7.22 ± 0.91
Mean glucose 30 (-30d-d0)	163.63 ± 38.21
GMI2 (-6w/-4w)	7.24 ± 0.93
Mean glucose2 (-6w/-4w)	164.13 ± 38.83
GMI3 (-10w/-8w)	7.25 ± 0.89
Mean glucose 3 (-10w/-8w)	164.51 ± 37.22

**Table 3 biosensors-12-00288-t003:** Correlation between GMI of selected time spans and HbA1c.

	R	*p*-Value
Correlation GMI1/HbA1c	0.82	<0.001
Correlation GMI30/HbA1c	0.84	<0.001
Correlation GMI2/HbA1c	0.80	<0.001
Correlation GMI3/HbA1c	0.78	<0.001

**Table 4 biosensors-12-00288-t004:** Linear regression analysis (CV: coefficient of variation; TiR: time in range; TaR: time above range; TbR: time below range).

	Estimate	95% CI (Upper/Lower Limit)	R-Squared	*p*-Value
Model statistic			0.1344	9.22 × 10^−5^
Intercept	2.415	8.38/13.21		
Age	0.004	−5.2 × 10^−5^/0.01		0.053
BMI	0.012	0.001/0.022		0.029
Sex	0.03	−0.086/0.146		0.61
Type 2 diabetes as factor	0.16	8.286 × 10^−4^/0.322		0.0488
Type 3 diabetes as factor	−0.02	−0.222/0.172		0.817
CV	0.875	−0.107/1.856		0.080
TiR	−0.028	−0.136/0.08		0.610
TaR	−0.028	−0.136/0.079		0.605
TbR	−0.027	−0.136/0.082		0.626
Absolute difference GMI1/GMI2	0.037	−0.16/0.234		0.71
Absolute difference GMI1/GMI3	0.218	0.059/0.378		0.0074

## Data Availability

Not applicable.

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
