# Peer review of "HbA1c and Glucose Management Indicator Discordance Associated with Obesity and Type 2 Diabetes in Intermittent Scanning Glucose Monitoring System"

_biosensors, 2022, doi:10.3390/bios12050288_

Round 1

Reviewer 1 Report

I enjoyed reading this paper. It’s a nice, sound study with an adequate design, and the findings are of interest. However, there are some comments I would ask the authors to address in their response and in the discussion part of their manuscript.

  1. Personally, as a diabetologist I do not use GMI a lot for treatment decision. It is evident that estimated HbA1c and laboratory HbA1c values can differ widely. Thus, in my analysis of patients’ CGM profiles I mainly focus on ‘time in range’, which seems to be a more appropriate marker of glycemic control. Thus, if CGM data is available, I would use TiR rather than GMI as a substitute for HbA1c (for TiR calculation based on DCCT data and correlation with diabetes related complications see Beck at al 2019 doi: 10.2337/dc18-1444). Could the authors comment on this and perhaps add a few sentences in their discussion?
  2. P 10, line 2010: ‘Previous studies concluded that a time span of ten to fourteen days of CGM data is 210 sufficient to estimate glycaemic control appropriately (Ref 3)’. In this context, this statement suggesting that this assumption might not be adequate is a bit misleading to me. The cited study did not look at GMI. It focused on mean glucose, time at 70–180 mg/dL, and time >180 mg/dL. Could the authors please clarify this statement.
  3. A recent review by Gomez-Peralta et al (doi 10.1111/dom.14638 ) very nicely summarizes the clinical implications of the differences between GMI and HbA1c in T1D. They readopt the concept of ‘high’ glycators (when HbA1c is consistently higher than GMI) and ‘low’glycators’ (when HbA1c is consistently lower than GMI) and the potential impact of this difference on glucose-mediated damage/diabetes-related complications. Perhaps the author could include this review in their discussion and speculate on the magnitude of the difference between GMI and HbA1c in any person with diabetes (particularly in T2D) as a marker of the risk of diabetes complications, particularly in type 2 diabetes along with the findings of this study.

Author Response

Thank you for the review and the great inputs. Below you find the point-by-point response.

 1.   Personally, as a diabetologist I do not use GMI a lot for treatment decision. It is evident that estimated HbA1c and laboratory HbA1c values can differ widely. Thus, in my analysis of patients’ CGM profiles I mainly focus on ‘time in range’, which seems to be a more appropriate marker of glycemic control. Thus, if CGM data is available, I would use TiR rather than GMI as a substitute for HbA1c (for TiR calculation based on DCCT data and correlation with diabetes related complications see Beck at al 2019 doi: 10.2337/dc18-1444). Could the authors comment on this and perhaps add a few sentences in their discussion?

  • A paragraph was added (see lines 289-292)

2.    P 10, line 2010: ‘Previous studies concluded that a time span of ten to fourteen days of CGM data is 210 sufficient to estimate glycaemic control appropriately (Ref 3)’. In this context, this statement suggesting that this assumption might not be adequate is a bit misleading to me. The cited study did not look at GMI. It focused on mean glucose, time at 70–180 mg/dL, and time >180 mg/dL. Could the authors please clarify this statement.

  • Changed the paragraph to clarify the statement (see lines 229-233)

3.    A recent review by Gomez-Peralta et al (doi 10.1111/dom.14638 ) very nicely summarizes the clinical implications of the differences between GMI and HbA1c in T1D. They readopt the concept of ‘high’ glycators (when HbA1c is consistently higher than GMI) and ‘low’glycators’ (when HbA1c is consistently lower than GMI) and the potential impact of this difference on glucose-mediated damage/diabetes-related complications. Perhaps the author could include this review in their discussion and speculate on the magnitude of the difference between GMI and HbA1c in any person with diabetes (particularly in T2D) as a marker of the risk of diabetes complications, particularly in type 2 diabetes along with the findings of this study.

- Thank you for this input. The review was added in the introduction and it is now also discussed briefly in the discussion (see lines 253-258)

Reviewer 2 Report

The manuscript takes a closer look at the relationship between obesity and diabetes (study patients were both T1D and T2D), which is known to be the most important modifiable risk for prevention of T2D. In the study, the authors aimed to investigate the glucose management indicator for predicting HbA1c in PWD's using the Abbott Freestyle Libre and to determine factors relating to deviations in measurements.

The study appears to be carried out very well, and the data backs up the conclusions found by the authors. The factors discussed by the authors, such as inflammation, time span of GMI calculation, and BMI are all accepted knowledge - the data they present adds to the existing knowledge of the community and is worthwhile for publication.

Experimental

  • Which version of the Freestyle Libre were the patients using? (Libre 1, Libre 2, Libre 3?)
  • Please give more detail regarding the statistical analysis completed in this study.

Grammar / Spelling

  • Figure 2 caption, the word ":measurements" at the very end appears to have been cut off
  • Figure 3, the words "Figure 3" should be bold.
  • In the figure captions, some figures use "Figure X:" and some use "Figure X." Please make uniform.

Author Response

Thank you for the review. Below you find the point-by-point response.

Experimental

    Which version of the Freestyle Libre were the patients using? (Libre 1, Libre 2, Libre 3?)

  • The population used primarily Freestyle Libre 1 as Freestyle Libre 2 was only started to be refunded by the Austrian health insurance, starting with end of 2020.

    Please give more detail regarding the statistical analysis completed in this study.

  • Part on statistical analysis was extended

Grammar / Spelling

    Figure 2 caption, the word ":measurements" at the very end appears to have been cut off

  • resolved

    Figure 3, the words "Figure 3" should be bold.

  • resolved

    In the figure captions, some figures use "Figure X:" and some use "Figure X." Please make uniform.

  • resolved

Reviewer 3 Report

In this work, authors investigated the accordance between GMI and HbA1c in patients with diabetes using intermittent scanning CGM 13
(isCGM). The number of patients used for the study are less but they have used one sensor type and longer observation periods that make the results more comparable.  The various results and tables show adequate analysis. The study has significance and can be accepted after minor revisions.

(1) At first, authors need to improve the English. At many places in the article, the tenses used are not accurate. Please revise it everywhere.

(2) There are several punctuations errors and spacing problems. Two words combined together at any places. Please check and improve.

(3) The introduction needs revision. I seen that only 8 references devoted for the introduction which is very less number. Either authors needs to add more references or cite rest of the references in the introduction.

(4) Many figure and table captions first word is small case letter. I think it should be capitalized for proper presentation.

(5) Fig. 2 (Bland plot) and Fig. 3 (absolute value) plots should be improved. The letter written inside the figures are too small. Please increase the font size.

(6) How many figures are there in the paper..? Authors have mentioned Fig. 1 two times in the captions. Please correct. 

(7) In Fig. 1 (Timeline of GMI), the number of weeks are shown using a minus or hyphen sign in front of numbers. Is it minus or hyphen..? I thing you should remove the signs. Its a positive number.

(8) In the section limitations and strengths, grammatical errors make it difficult to understand to meaning. For example, "In conclusion, there is a great and often clinically relevant difference between GMI 259 and HbA1c when using isCGM". 

(9) The title needs bit improvement. Writing word "is" makes the title looks like a sentence. Please improve it.

Author Response

Thank you for the review. Below you find the point-by-point response.

In this work, authors investigated the accordance between GMI and HbA1c in patients with diabetes using intermittent scanning CGM (isCGM). The number of patients used for the study are less but they have used one sensor type and longer observation periods that make the results more comparable.  The various results and tables show adequate analysis. The study has significance and can be accepted after minor revisions.

(1) At first, authors need to improve the English. At many places in the article, the tenses used are not accurate. Please revise it everywhere.

  • revised

(2) There are several punctuations errors and spacing problems. Two words combined together at any places. Please check and improve.

  • revised

(3) The introduction needs revision. I seen that only 8 references devoted for the introduction which is very less number. Either authors needs to add more references or cite rest of the references in the introduction.

  • More references were added

(4) Many figure and table captions first word is small case letter. I think it should be capitalized for proper presentation.

  • resolved

(5) Fig. 2 (Bland plot) and Fig. 3 (absolute value) plots should be improved. The letter written inside the figures are too small. Please increase the font size.

  • Increased font size of Fig 2-4.

(6) How many figures are there in the paper..? Authors have mentioned Fig. 1 two times in the captions. Please correct.

  • There are 4 figures in the paper. The captions are correct now

(7) In Fig. 1 (Timeline of GMI), the number of weeks are shown using a minus or hyphen sign in front of numbers. Is it minus or hyphen..? I thing you should remove the signs. Its a positive number.

  • revised

(8) In the section limitations and strengths, grammatical errors make it difficult to understand to meaning. For example, "In conclusion, there is a great and often clinically relevant difference between GMI 259 and HbA1c when using isCGM".

  • revised the sentence

(9) The title needs bit improvement. Writing word "is" makes the title looks like a sentence. Please improve it.

  • - revised